# Detection of serum IgG autoantibodies to FcεRIα by ELISA in patients with chronic spontaneous urticaria

**Jae-Hyuk Jang**, **Jiyoung Moon, Eun-Mi Yang, Min Sook Ryu, Youngsoo Lee, Young-Min Ye, Hae-Sim Park**\*

Department of Allergy and Clinical Immunology, Ajou University School of Medicine, Suwon, Korea

\* hspark@ajou.ac.kr

**Data Availability Statement:** All relevant data are within the paper and its Supporting information files.

## Abstract

### Background

Mast cells are a key effector cell in the pathogenesis of chronic spontaneous urticaria (CSU) and activated by circulating FcεRI-specific IgG as well as IgE. This study evaluated the prevalence of circulating autoantibodies to FcεRIα in the sera of CSU patients.

### Methods

Eighty-eight patients with CSU and 76 healthy controls (HCs) were enrolled. To detect circulating autoantibodies (IgG/IgA/IgM) to FcεRIα, ELISA was done using YH35324 (as a solid phase antigen), and its binding specificity was confirmed by the ELISA inhibition test. The antibody levels were presented by the ratio of YH35324-preincubated to mock-preincubated absorbance values. Clinical and autoimmune parameters, including atopy, urticaria activity score (UAS), serum total/free IgE levels, serum antinuclear antibody (ANA) and autologous serum skin test (ASST) results, were assessed. The autoimmune group was defined if CSU patients had positive results to ASST and/or ANA.

### Results

The ratio of serum IgG to FcεRIα was significantly lower in CSU patients than in HCs (*P*<0.05), while no differences were noted in serum levels of IgG to recombinant FcεRIα or IgA/IgM autoantibodies. The autoimmune CSU group had significantly lower ratios of IgG/IgA (not IgM) autoantibodies to FcεRIα than the nonautoimmune CSU group (*P*<0.05 for each). No significant associations were found between sex, age, atopy, urticaria duration, UAS, or serum total/free IgE levels according to the presence of IgG/IgA/IgM antibodies.

### Conclusions

This study confirmed the presence of IgG to FcεRIα in the sera of CSU patients, especially those with the autoimmune phenotype.

**Funding:** This study was supported by a grant of the Korea Health Technology R&D Project (HR16C0001) through the Korea Health Industry Development Institute, funded by the Ministry of Health & Welfare, Republic of Korea, and partly supported and provided with YH35324 by Yuhan, Seoul, Korea. The funders had no role in study design, data collection and analysis, or preparation of the manuscript.

**Competing interests:** The authors have declared that no competing interests exist.

## Introduction

Chronic spontaneous urticaria (CSU) is defined as the occurrence of wheals, angioedema, or both for more than 6 weeks due to unknown causes [1]. The prevalence rate of CSU was reported at 0.5%-5% in the general population and has been increasing worldwide along with industrialization [2, 3].

Mast cell is a key effector cell, and when activated by various triggers, it releases mediators and cytokines like histamine. Recent guidelines recommend the measurement of serum auto-antibody, IgG to thyroid peroxidase (TPO), which is related to not only thyroid diseases but also autoimmune diseases, to evaluate autoimmunity in patients with CSU [1, 4].

Although IgE-mediated mechanism is a major one to activate mast cells, recent studies demonstrate autoimmune-mediated mechanisms in patients with CSU, especially those with more severe phenotypes [5–7]. A type IIb hypersensitivity as an autoantibody-mediated mechanism was first reported in CSU [8]. A-positive result to the autologous serum skin test (ASST) was related to the presence of autoantibodies in CSU [9]. However, the prevalence or role of each autoantibody in CSU has not been fully understood [10, 11]. Among autoantibodies, IgG to FcεRIα has been reported to be a major autoantibody for activating mast cells in CSU patients; however, a recent report has demonstrated a higher prevalence of IgM to FcεRIα (than IgA and IgG antibodies to FcεRIα) in the sera of CSU patients [11].

Regarding the detection method of IgG autoantibody to FcεRIα, several different methods were reported; a standardized method is still unavailable. We could measure serum free IgE using YH35324 (capturing the FcεRIα part of free IgE) in the sera of asthmatics [12] and hypothesized that this YH35324 could capture the FcεRIα binding site of FcεRIα-specific anti-bodies. Then, we extended ELISA to detect circulating IgG/IgA/IgM autoantibodies to FcεRIα in the sera of CSU patients compared to controls. In the present study, we aimed to evaluate the prevalence of serum IgG/IgA/IgM autoantibodies to FcεRIα in association with clinical/autoimmune parameters in patients with CSU.

## Methods

### Study subjects

The present study enrolled 88 CSU patients and 76 healthy controls (HCs) from the Department of Allergy and Clinical Immunology, Ajou University Hospital (Suwon, South Korea). Patients who had urticaria symptoms for over 6 weeks without any inducible causes were diagnosed as having CSU. HCs had no history of allergic diseases like urticaria or autoimmune disease.

CSU patients were over 18 years old and had never been exposed to IgE-related biologics. They were allowed to use control medications including antihistamines, except steroids, for 4 weeks prior to the study, and their sera were collected at the diagnosis and stored at −80˚C until estimation. The levels of serum IgG/IgA/IgM autoantibodies to FcεRIα were measured by ELISA with applying YH35324 and compared between CSU patients and HCs. All participants provided written informed consent and agreed to voluntarily participate in the study. This study was approved by the Institutional Review Board of Ajou University Hospital (AJIRB-BMR-SMP-20-435).

The demographic and clinical characteristics of the study subjects were obtained by the investigators. The severity of CSU including disease duration, current medication requirements, and the urticaria activity score (UAS-15) were analyzed. The UAS-15 included the degree of pruritus and the number/size/distribution/duration of wheal over the preceding week at the initial visit. Patients were classified according to the severity of symptoms as

follows: 0, symptom-free; 1 to 5, mild urticaria; 6 to 10, moderate urticaria; and 11 to 15, severe urticaria. Concomitant allergic disease and atopic status were documented.

Skin prick tests (SPT) were performed with 55 common aeroallergens (Bencard, Brentford, UK), which included *Dermatophagoides pteronyssinus*, *D. farinae*, histamine, and a saline negative control. A positive reaction was defined as a wheal of more than 3 mm in diameter or wheal size larger than a positive control (histamine) after 15 minutes. If SPT was not possible, specific IgE antibodies to clinically relevant allergens, including house dust mite (HDM)-specific IgE, were measured. Atopy was defined when SPT showed a positive results or serum allergen specific IgE was elevated ($\geq$0.35 IU/mL) to at least 1 common environmental allergen. The levels of serum total and specific IgE were measured by ImmunoCAPs (ThermoFisher Scientific, Waltham, MA, USA). Serum free IgE level was measured by ELISA as previously described [12].

## Evaluation of autoimmunity in CSU

Routine laboratory measurements were taken and included complete blood count, liver enzymes, thyroid stimulating hormone (TSH), and the antinuclear antibody (ANA) screening test. In the cases of abnormal TSH results, thyroid autoantibodies (anti-TPO or anti-TG) were checked. The autologous serum skin test (ASST) was performed to evaluate autoimmune characteristics in CSU patients. The subjects who underwent the ASST had an intradermal injection of 50 μL of their own sera into their arm. After 30 minutes, the patient's wheal and flare responses were compared with the responses to normal saline and histamine as controls. An induced wheal larger than 1.5 mm was considered a positive result [13].

## ELISA for the measurement of serum autoantibodies

The levels of serum autoantibodies to FcɛRIα were measured using ELISA by using YH35324 (Yuhan, Seoul, South Korea). This novel YH35324 is based on a human IgG4 Fc domain combined with a human FcɛRIα extracellular domain on its Fab chain. As a FcɛRIα domain, we speculated that YH35324 could have not only high affinity for serum free IgE but also circulating IgG autoantibodies to FcɛRIα. Based on its affinity, we developed a new ELISA using this YH35324 for the measurement of IgG/IgA/IgM autoantibodies to FcɛRIα. In brief, the YH35324 in bicarbonate buffer (pH 9.6) was coated at 1μg/mL onto each well (a solid phase) overnight at 4˚C. After washing 3 times with 0.05% Tween 20 containing phosphate-buffered saline (PBST), the plates were blocked with buffer (10% FBS containing PBST) for 1 hour.

Sera from CSU and HCs were pre-incubated for 1 hour with 5 μg/mL YH35324 or mock (10% FBS containing PBST). Then, 2 kinds of sera (pre-incubated with YH35324 or mock at 1:10 dilution) were loaden onto antigen-coated wells at room temperature for 2.5 hours and washed 5 times with PBST. After washing, each well was incubated with horseradish peroxidase (HRP)-conjugated secondary antibody at room temperature and washed 5 times with PBST.

For the measurement of IgG autoantibodies to FcɛRIα, purified serum by using the Gel-IgG-Spin purification kit (ThermoFisher Scientific, Rockford, IL, USA) was incubated, and HRP-conjugated anti-IgG antibody (1:2000, BD Biosciences, Franklin Lakes, NJ, USA) was used as a second antibody, followed by 2 hours of incubation. In the case of IgA and IgM autoantibodies to FcɛRIα, sera purification was not applied. HRP-conjugated anti-human IgA and anti-human IgM (1:3000 for IgA, 1:2000 for IgM, SouthernBiotech, Birmingham, AL, USA) were used as second antibodies and incubated for 1.5 hours.

Finally, the plates were incubated with tetramethylbenzidine (TMB) substrate (BD Biosciences) for 20 minutes, followed by stopping the reaction with 2N $H_2SO_4$. The absorbance

values were read using an ELISA reader (BioTek, Winooski, VT, USA) at 450 and 570 nm, and the subtracted values were analyzed. All samples were measured 3 times repeatedly, and mean values were analyzed after removing outliers. The coefficient of variation in the reproducibility of ELISA for IgG to FcεRIα was 13.7%. The serum levels of IgG to recombinant FcεRIα were measured simultaneously and compared between patients with CSU and HCs according to the same protocol for the measurement of IgG to FcεRIα except that each well was coated with recombinant soluble FcεRIα (MyBioSource, San Diego, CA, USA) as a solid phase antigen and recombinant soluble FcεRIα was used as an inhibitor instead YH35324.

The levels of IgG/IgA/IgM autoantibody were presented as the ratio of YH35324 preincubated to mock-preincubated values. Reduced levels after the pre-incubation was calculated, and lower ratios represent higher levels of circulating autoantibodies [11].

## ELISA inhibition test

Competitive ELISA inhibition tests were performed to confirm the binding specificity of YH35324. Sera from patients with CSU were incubated overnight at 4˚C, with increasing amounts (0–1000 μg/mL protein concentration) of YH35324 before use. The pretreated sample was incubated with the YH35324-coated microtiter plate, and ELISA was carried out as previously described. *D. farinae* was used as a nonspecific allergen. The inhibition rate was calculated by the formula: inhibition rate (%) = 100 –(absorbance with inhibitor/absorbance without inhibitor) x100.

## Statistical analysis

To evaluate the association among 3 autoantibodies and clinical parameters, parametric and non-parametric continuous variables were compared using Student's *t test* and the Mann-Whitney test. Categorical values were compared Pearson's Chi-squared test and Fisher's exact test. Parametric and non-parametric variables were compared among over 2 groups using ANOVA and the Kruskal-Wallis test. If standard deviations were not equal in parametric variables, Welch's ANOVA test was used. Bonferroni and Dunnett were applied as *post hoc* tests. Correlation coefficients were calculated by the Pearson method. Cutoff values for the prevalence of circulation autoantibodies to FcεRIα were determined at the 5% percentile ratio of non-autoimmune CSU. All statistical analyses were performed, and graphs were created using R 4.1.0 (R Core Team, 2021), GraphPad Prism Version 9.2.0 for Mac (GraphPad Software, San Diego, CA, USA). Statistical significance was set at *P* less than 0.05 for all tests.

## Results

### Clinical characteristics of the study subjects

Table 1 shows the baseline characteristics of patients enrolled compared to HCs. Patients with CSU was younger than HCs (median 37.5 [IQR 25.0–48.0] *vs*. median 41.0 [IQR 34.0–52.0], $P = 0.01$). No differences were found in sex or atopy prevalence. The mean UAS was 9.2±4.3; positive rates of ASST and ANA were 22.7% and 18.2%, respectively. Both serum total and free IgE levels were significantly higher in CSU patients than in HCs, with significant differences ($P<0.001$ for each). Two patients with CSU had hyperthyroidism and 1 patient with CSU had hypothyroidism. Eight patients with abnormal results of TSH or a history of thyroid disease were positive to at least one thyroid autoantibody (anti-TPO or anti-TG). Patients with positive thyroid autoantibodies did not have any autoantibodies to FcεRIα. S1 Fig showed the correlation between IgG to YH35324 and IgG to recombinant FcεRI. (*Pearson r* = 0.41, $P<0.0001$).

**Table 1. Clinical characteristics of the study subjects in the cohort.**

| Variable | CSU | Healthy controls | P value |
|---|---|---|---|
| | (n = 88) | (n = 77) | |
| Female, n (%) | 54 (61.4%) | 39 (50.6%) | 0.22 |
| Age (years) | 37.5 [25.0;48.0] | 41.0 [34.0;52.0] | 0.01 |
| Atopy, n (%) | 54 (61.4%) | 36 (50.0%) | 0.20 |
| UAS-15 | 9.2±4.3 | NA | |
| ASST positive, n (%) | 20 (22.7%) | NA | |
| ANA positive, n (%) | 16 (18.2%) | NA | |
| Serum total IgE (kU/L) | 135.5 [64.5;312.5] | 50.7 [20.4;110.5] | <0.001 |
| Serum free IgE (ng/mL) | 252.9 [99.5;639.8] | 105.9 [23.4;273.5] | <0.001 |

Non-parametric values are presented as median [IQR] and parametric values or mean ± SD. Categorical values are shown as number (percentage). *P* values were evaluated by the *t* test or the Mann-Whitney test. CSU, chronic spontaneous urticaria; HCs, healthy controls; UAS, urticaria activity score; ASST, autologous serum skin test; ANA, antinuclear antibody, NA: not available

## The prevalence of serum autoantibodies (IgG/IgA/IgM) to FcεRIα in CSU patients

Fig 1 shows the comparison of serum IgG/IgA/IgM levels to FcεRIα between patients with CSU and HCs. The ratio of IgG to FcεRIα was significantly lower in patients with CSU than in HCs, while no differences were noted in the ratio of IgG to recombinant FcεRIα or IgA/IgM autoantibodies to FcεRIα. The ELISA inhibition test confirmed binding specificity by showing a dose-dependently increasing inhibition rate (%) with serial addition of the YH35324 (0–1000 μg/mL) in the sera of 1 patient with CSU (S2 Fig).

Table 2 summarizes comparisons of clinical parameters with the results of serum IgG/IgA/IgM to FcεRIα. The cutoff value for the positive predictive value of IgG/IgA/IgM to FcεRIα was determined from the results of non-autoimmune CSU and shown as dotted lines (0.65, 0.72, and 0.71, respectively, Figs 1 and 2). The prevalence of serum IgG/IgA/IgM to FcεRIα in CSU were 8.1, 5.8, and 3.4% respectively. No significant differences were noted in demographic findings, clinical parameters (sex, age, atopic status, disease duration, UAS, and serum total/free IgE), or comorbid conditions (allergic rhinitis and asthma) according to the presence of serum IgG/IgA/IgM to FcεRIα. In addition, no differences were noted in peripheral eosinophil or basophil counts between the groups with positive and negative to IgG/IgA/IgM autoantibody. The levels of HDM-specific IgE were higher in the IgG-positive group than in the IgG-negative group (median values of *D. pteronyssinus*-specific IgE, 18.1 [IQR 1.1–24.9] kU/L *vs.* 0.1 [IQR 0.1–1.8] kU/L, *P*<0.05; the median values of *D. farinae*-specific IgE, 33.9 [IQR 1.3–45.7] kU/L *vs.* 0.3 [IQR 0.1–2.5] kU/L, *P*<0.05), while no significant associations were found with atopic status or serum IgE levels between the 2 groups. No differences were found in the results of IgA/IgM autoantibodies as shown in Table 2.

## Comparison of IgG to FcεRIα levels with the results of ASST and ANA

When defined if CSU patients had at least 1 positive result to ASST or ANA, the autoimmune CSU group had significantly lower ratio of IgG/IgA to FcεRIα than the non-autoimmune CSU group (the median values of IgG to FcεRIα, 0.73 [IQR 0.68–0.81] *vs.* 0.78 [IQR 0.72–0.89], *P*<0.05; the median values of IgA to FcεRIα, 0.93 [IQR 0.87–1.01] *vs.* 0.99 [IQR 0.93–1.05]) (*P*<0.05, Fig 2). IgM to FcεRIα showed no difference between the 2 groups. Comparison of

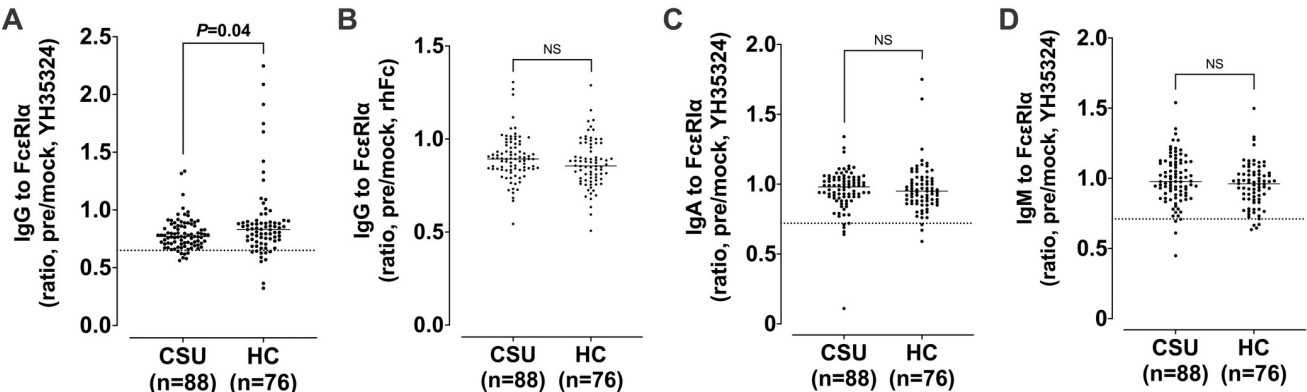

**Fig 1. Comparison of serum IgG, IgA, and IgM autoantibodies to FcεRIα between patients with chronic spontaneous urticaria (CSU) and healthy controls (HCs).** The results were presented by the IgG ratio of YH35324-pretreated to mock-treated values (A), the IgG ratio of rhFc-pretreated to mock-treated value (B), the IgA ratio of YH35324-pretreated to mock-treated value (C) and the IgM ratio of YH35324-pretreated to mock-treated value (D). Horizontal lines and dotted lines show median values and cutoff values of each autoantibody to FcεRIα respectively. P values were evaluated by Mann-Whitney test. rhFc, recombinant human Fc; NS, no statistical significance.

serum autoantibodies between the ASST-positive and ASST-negative groups showed no significant difference (S3 Fig).

## Discussion

The present study demonstrates the presence of circulating IgG antibody to FcεRIα by the home-made ELISA using purified IgG sera and the YH35324 (capturing the FcεRIα portion of specific IgG to FcεRIα) as the solid phase antigen. The IgG-ELISA inhibition test

**Table 2. Clinical characteristics of the patients with chronic spontaneous urticaria (CSU) according to the results of serum autoantibodies to FcεRIα.**

| Variables (n = 86) | IgG to FcεRIα | | IgA to FcεRIα | | IgM to FcεRIα | |
|---|---|---|---|---|---|---|
| | Negative | Positive | Negative | Positive | Negative | Positive |
| | (n = 79, 91.8%) | (n = 7, 8.1%) | (n = 81, 94.1%) | (n = 5, 5.8%) | (n = 83, 96.5%) | (n = 3, 3.4%) |
| Female, n (%) | 50 (63.3%) | 3 (42.9%) | 49 (60.5%) | 4 (80.0%) | 51 (61.4%) | 2 (66.7%) |
| Age (years) | 37.0 [25.0;48.5] | 44.0 [38.0;47.0] | 38.0 [27.0;48.0] | 22.0 [22.0;42.0] | 38.0 [25.0;48.0] | 47.0 [41.0;51.5] |
| Angioedema, n (%) | 38 (48.1%) | 3 (42.9%) | 39 (48.1%) | 2 (40.0%) | 40 (48.2%) | 1 (33.3%) |
| UAS-15 | 10.0 [6.0;13.0] | 7.0 [5.5;11.5] | 10.0 [6.0;13.0] | 9.0 [2.0;12.0] | 10.0 [6.0;13.0] | 15.0 [9.0;15.0] |
| Duration (months) | 10.0 [2.5;27.0] | 7.0 [4.5;8.0] | 8.0 [2.5;24.0] | 10.0 [3.0;12.0] | 9.0 [3.0;24.0] | 1.0 [0.6;3.0] |
| ASST positive, n (%) | 17 (21.5%) | 3 (42.9%) | 19 (23.5%) | 1 (20.0%) | 20 (24.1%) | 0 (0.0%) |
| ANA positive, n (%) | 13 (16.5%) | 3 (42.9%) | 15 (18.5%) | 1 (20.0%) | 15 (18.1%) | 1 (33.3%) |
| Serum total IgE (kU/L) | 132.0 [59.5;288.0] | 396.0 [179.5;538.0] | 139.0 [61.0;294.0] | 182.0 [93.0;520.0] | 143.0 [65.5;312.5] | 68.0 [37.0;296.5] |
| Serum free IgE (ng/mL) | 254.8 [100.7;555.3] | 788.1 [129.1;797.8] | 254.8 [99.3;611.8] | 311.4 [250.9;781.1] | 266.7 [100.7;639.8] | 163.2 [83.3;478.3] |
| Atopy, n (%) | 46 (58.2%) | 7 (100.0%) | 48 (59.3%) | 5 (100.0%) | 52 (62.7%) | 1 (33.3%) |
| *Der p* IgE (kU/L) | 0.1 [0.1;1.8] * | 18.1 [1.1;24.9] * | 0.2 [0.1;2.4] | 0.5 [0.2;2.0] | 0.2 [0.1;2.0] | 0.2 [0.1;3.4] |
| *Der f* IgE (kU/L) | 0.3 [0.1;2.5] * | 33.9 [1.3;45.7] * | 0.4 [0.1;4.3] | 1.4 [0.2;1.8] | 0.4 [0.1;3.9] | 0.2 [0.2;3.0] |
| Eosinophil counts (cells/μL) | 93.5 [57.6;161.8] | 62.4 [44.9;155.1] | 93.5 [57.6;162.8] | 65.0 [36.0;156.0] | 93.5 [55.2;168.9] | 70.4 [35.2;96.0] |
| Basophil counts (cells/μL) | 32.0 [22.8;43.0] | 39.0 [33.1;54.8] | 32.4 [22.8;44.0] | 39.6 [32.5;42.7] | 33.0 [23.7;43.6] | 22.8 [14.6;47.4] |

Non-parametric values are shown as median [IQR] and categorical values are shown as number (percentage).

*P* values were evaluated by Pearson's Chi-squared test or the Mann-Whitney test.

*$P<0.05$, between patients with negative results and those with positive results to IgG to FcεRIα.

UAS, urticaria activity score; ASST, autologous serum skin test; ANA, antinuclear antibody; *Der p*, *Dermatophagoides pteronyssinus*; *Der f*, *Dermatophagoides farinae*

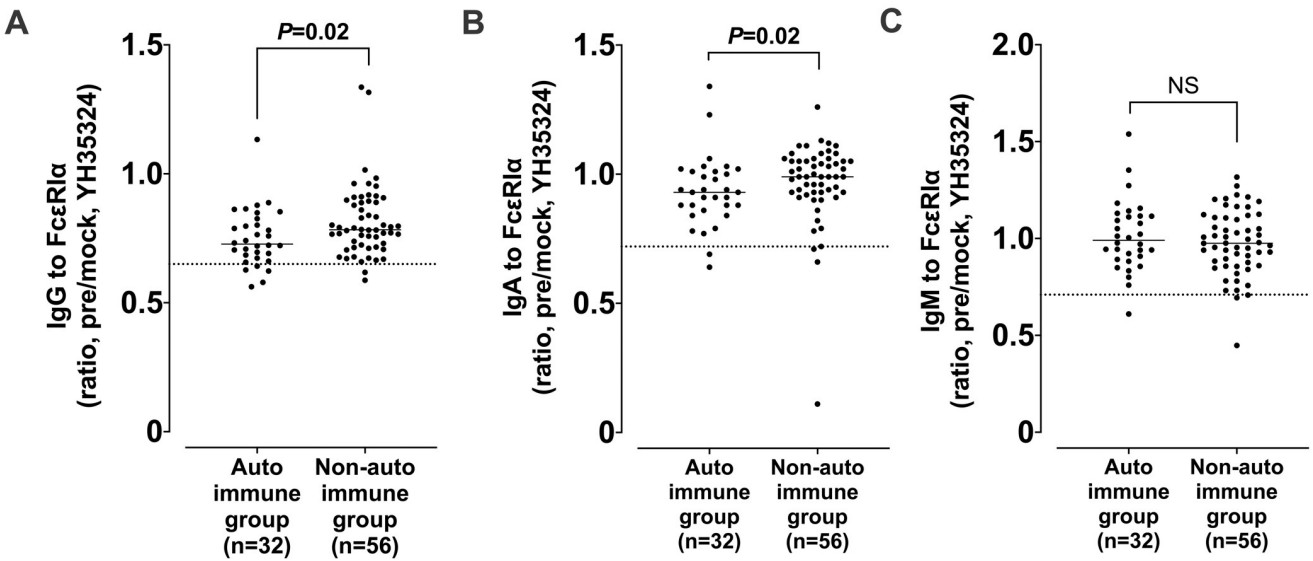

**Fig 2. Comparisons of serum IgG/IgA/IgM levels to FcεRIα between the autoimmune and nonautoimmune groups.** The results were presented by the IgG ratio of YH35324-pretreated to mock-treated value (A), the IgA ratio of YH35324-pretreated to mock-treated value (B) and the IgM ratio of YH35324-pretreated to mock-treated value (C). The autoimmune group was defined if CSU patients have a positive result to ASST or ANA test. Horizontal lines and dotted lines show median values and cutoff values of each autoantibody FcεRIα respectively. P value was evaluated by Mann-Whitney test. ASST, autologous serum skin test; ANA, antinuclear antibody; NS, no statistical significance.

demonstrated the binding specificity with serial additions of YH35324 (up to 88.85% inhibition). In addition, higher levels of serum IgG/IgA to FcεRIα were noted in patients with CSU with autoimmune phenotypes, while serum IgG to recombinant FcεRIα were not detectable in the same serum set of the study subjects. These findings suggest that our ELISA is a reproducible and reliable method to detect serum IgG level to FcεRIα in the sera of various allergic and immunologic diseases, although further replication studies are needed.

For CSU, the autoimmune mechanism plays a major role in activating mast cells [14, 15]. CSU patients with type IIb autoimmunity have serum autoantibodies to FcεRIα or IgE on the surface of mast cells [11, 16]. From the first study on IgG to FcεRIα in patients with CSU to the latest study evaluating the prevalence of IgG, IgA and IgM to FcεRIα in those with CSU, autoimmune CSU has long been studied [11, 17]. However, there are no standardized method to measure serum IgG/IgA/IgM autoantibodies to FcεRIα, and the reference values for determining autoimmunity are not determined.

To define type IIb autoimmune CSU, 3 diagnostic criteria have been recommended: (1) positive ASST, (2) positive basophil histamine release assay and/or basophil activation test, and (3) positive IgG autoantibodies against FcεRIα and/or IgE [4, 7]. CSU patients with positive ASST results present more severe symptoms, including wheals, itching, and systemic manifestations, than those with negative ASST results [7]. However, no difference was noted in this study. The positive rate of the ASST was 22.7% in the present study, which is lower than those of previous studies (34% to 70%) [18, 19]. Autoimmune thyroiditis was reported to be related to the persistent positive result of ASST in patients with CSU independent of urticaria symptoms, questioning the role of ASST in the CSU patients with autoimmune thyroiditis [20]. Further long-term follow-up studies are needed to evaluate its clinical relevance (effects on drug responses or long-term clinical outcomes) in the management of CSU.

The ASST has been increasingly performed and revealed that it is associated with IgG antibodies against the FcεRIα or IgE. The positive reaction in the ASST was reported to confirm

the presence of autoantibodies in CSU [9]. A previous study showed higher serum IgG to FcεRIα estimated by immunoblot assay in CSU patients, especially in those with a positive ASST, although further replication studies are not validated [10]. These findings suggest that IgG to FcεRIα may be involved in autoantibody-mediated mechanism of CSU. However, the present study showed no significant difference in IgG to FcεRIα between the ASST-positive and ASST-negative group (S3 Fig). The discordance between previous studies and ours in the same ethnicity may be attributed to: (1) differences in the positive rates of ASST found in the study subjects and (2) the different detection methods applied (immune-dot assay *vs.* ELISA). The sensitivity and specificity between these 2 different assays cannot be comparable [21].

Regarding autoimmune parameters in CSU, ANA has commonly been reported to be present in CSU patients, especially those associated with the autoimmunity type IIb [22]. The ANA-positive group is known to be associated with poor responses to antihistamines or omalizumab [23, 24]. The positive ANA rate in CSU patients were 18.2% in the present study, which is consistent with the positive rates (15%-29%) of previous studies [10, 23, 25]. However, no differences were found in UAS or other clinical parameters between the ANA-positive and ANA-negative groups, similar to a previous study [24].

To evaluate autoimmunity in CSU, basophil autoreactivity functional assay is needed, but it is difficult to apply in every patient in the aspects of technical difficulties, cost, and low availability in the real-world practice [26]. Therefore, basophil histamine release assay and/or basophil activation test were not widely performed in real-world study or practice.

The prevalence of IgG autoantibody to FcεRIα was reported from 4% to 64% in CSU patients [27]. For IgM and IgA autoantibodies, there has been a study reporting higher prevalence (60% and 57%, respectively) in CSU patients [11]. The present study showed a higher prevalence of IgG to FcεRIα in CSU patients than in HCs, but no differences were noted in the prevalence of IgM or IgA autoantibodies to FcεRIα between CSU patients and controls. In addition, when they were compared according to autoimmunity, the autoimmune group had a higher prevalence of IgG to FcεRIα than the nonautoimmune group, although the prevalence of IgG to FcεRIα was 8.1%, which were lower than those of previous studies [11]. The wide range of the prevalence of autoantibodies in CSU has not yet been clearly explained. There are a few explanations of these differences. First, the measurement methods were different among the studies, where western blot analysis, ELISA (with using different antigens/antibodies) or both assays were applied [27]. Further studies are required to confirm the results. Secondly, the discrepancies in autoantibody prevalence may have been attributed to differences in underlying causes and regional differences [1]. Even using the same measurement method, the prevalence rates were variable [10]. Thirdly, the cutoff criteria were different whether they were derived from HCs or other control subjects [10, 11]. The number of the study subjects having high levels of serum autoantibodies to FcεRIα was very low, and some of HCs had this autoantibody without any evidence of autoimmune disease or CSU. Therefore, the cutoff values from HCs may have failed to differentiate the autoimmune CSU group from the non-autoimmune CSU group. It is also supported by the fact that the prevalence of disease-related autoantibodies in the disease-free subgroup from the European cohort of autoantibody-related disease was found to be 23.6% [28].

There have been several studies showing that the presence of IgG autoantibodies against FcεRIα is linked to more severe symptoms, a poor response to conventional antihistamine treatment, and a slower response to anti-IgE treatment [11, 29, 30]. IgM to FcεRIα was reported to have a correlation with peripheral basopenia and eosinopenia [11]. IgA to FcεRIα was related to present or past gastrointestinal-or mucosal-associated infections [11]. The present study showed no significant associations between IgG to FcεRIα and clinical/severity parameters. Recent studies suggest that, along with mast cells, eosinophils, basophils and other

cells are involved in CSU [31]. Both lesional or non-lesional skin of CSU patients showed a mixed infiltrating cells such as eosinophils and basophils [1, 31–33]. Recruitment of eosinophils into the skin can be mediated by the interactions between mast cells and various mediators. Eosinophil granule proteins from activated eosinophils cause persistent mast cell activation, and IgG-anti-FcεRII/CD23 leads to eosinophil secretion via the low-affinity IgE receptor FcεRII/CD23 on eosinophils and histamine release from basophils, suggesting the role of interactions between mast cells and basophils/eosinophils in the pathogenesis of CSU. When peripheral eosinophil and basophil counts were compared according to the presence of serum IgG to FcεRIα, no differences were found in the present study, indicating that their immune cells are not associated with IgG autoantibody to FcεRIα in CSU.

There are 2 limitations in this study. One is that the operational definition of autoimmune CSU without basophil functional assay may be insufficient to reflect real autoimmunity. Basophil functional assays are the only currently available method to evaluate functional autoantibodies, however they are difficult to conduct in outpatient clinics due to technical difficulties and costs. Both tests for ANA and ASST are poor surrogate markers for a risk of false positivity and a possibility of the co-existence of atopic status [13, 34]. Nonetheless, they are potential alternatives to evaluate autoimmunity which is clinically relevant in CSU patients [13, 23, 35]. The other is that we could not find any clinical significance according to the presence of IgG to FcεRIα as well as IgA/IgM autoantibodies to FcεRIα, which may have been attributed to a lower number of positive responders. Further replication studies in a larger cohort of CSU patients are needed to validate these findings.

In conclusion, we detected higher IgG autoantibody levels to FcεRIα by the ELISA with applying the YH35324 in the sera of CSU patients compared to HCs, which were higher in the autoimmune group, suggesting a possible involvement of circulating autoantibodies against FcεRIα in the autoimmune mechanism of CSU.

## Supporting information

**S1 Fig. Correlation between log transformed IgG to YH35324 and log transformed IgG to recombinant FcεRI before pretreatment analyzed by Pearson's correlation.**
(TIF)

**S2 Fig. Competitive ELISA inhibition tests with serial additions of YH35324 and *D*. farina extracts (200μg/mL).** ELISA, enzyme-linked immunosorbent assay; *D*. farinae; *Dermatophagoides farinae.*
(TIF)

**S3 Fig. Comparison of serum IgG/IgA/IgM levels to FcεRIα between the ASST-positive and ASST-negative group.** The results were presented by the IgG ratio of YH35324-pretreated to mock-treated value (A), the IgA ratio of YH35324-pretreated to mock-treated value (B) and the IgM ratio of YH35324-pretreated to mock-treated value (C). ASST, autologous serum skin test; NS, no statistical significance.
(TIF)

## Author Contributions

**Conceptualization:** Hae-Sim Park.

**Data curation:** Jae-Hyuk Jang, Jiyoung Moon, Eun-Mi Yang, Min Sook Ryu, Youngsoo Lee.

**Formal analysis:** Jae-Hyuk Jang, Eun-Mi Yang.

**Investigation:** Jiyoung Moon.

**Methodology:** Eun-Mi Yang, Min Sook Ryu, Youngsoo Lee, Young-Min Ye.

**Supervision:** Young-Min Ye, Hae-Sim Park.

**Visualization:** Jae-Hyuk Jang.

**Writing – original draft:** Jae-Hyuk Jang.

**Writing – review & editing:** Hae-Sim Park.

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
