## [Decision Letter · Decision Letter 0]

4 Jul 2022

PONE-D-22-12472Detection of serum IgG autoantibodies to FcεRIα by ELISA in patients with chronic spontaneous urticariaPLOS ONE

Dear Dr. Park,

Thank you for submitting your manuscript to PLOS ONE. After careful consideration, we feel that it has merit but does not fully meet PLOS ONE’s publication criteria as it currently stands. Therefore, we invite you to submit a revised version of the manuscript that addresses the points raised during the review process.

We look forward to receiving your revised manuscript.

Kind regards,

Cheorl-Ho Kim, Ph.D.

Academic Editor

PLOS ONE

Journal Requirements:

3. Please upload a copy of Figures 1 and 2.

Additional Editor Comments :

July 4, 2022

Dear Dr. Hae-Sim Park

Ref: PONE-D-22-12472

Title: Detection of serum IgG autoantibodies to FcεRIα by ELISA in patients with chronic spontaneous urticaria

Journal: PLOS ONE

Thank you for your choosing us for your publication medium. Your study on the autoantibodies to the IgE receptor is interesting in our readers. However, there are some demerits in the discrete evidences and setting of experimental observation.

We have completed the review process of your manuscript. As you can read the attached comments, there are several contriversial issues in your study, causing for the 6 independent reviews, which are not general.

One reviewer is very positive, but the other criticisms are so important for your research to justify.

I believe that the comments can help you to revise your MS.

I hope you can easily revise it.

Thank you

Sincerely,

Cheorl-Ho Kim Prof

Sungkyunkwan Univ, Biol Science Dept.

Editor of Plos One

Reviewers' comments:

Reviewer's Responses to Questions

**Comments to the Author**

1. Is the manuscript technically sound, and do the data support the conclusions?

Reviewer #1: Yes

Reviewer #2: Partly

Reviewer #3: No

Reviewer #4: Partly

Reviewer #5: No

Reviewer #6: Yes

2. Has the statistical analysis been performed appropriately and rigorously? 

Reviewer #1: Yes

Reviewer #2: Yes

Reviewer #3: Yes

Reviewer #4: I Don't Know

Reviewer #5: Yes

Reviewer #6: Yes

3. Have the authors made all data underlying the findings in their manuscript fully available?

Reviewer #1: Yes

Reviewer #2: Yes

Reviewer #3: Yes

Reviewer #4: No

Reviewer #5: Yes

Reviewer #6: No

4. Is the manuscript presented in an intelligible fashion and written in standard English?

Reviewer #1: Yes

Reviewer #2: Yes

Reviewer #3: No

Reviewer #4: No

Reviewer #5: No

Reviewer #6: Yes

5. Review Comments to the Author

Reviewer #1: The objective of this study was to investigate the prevalence of circulating autoantibodies to FcεRIα in association with clinical/autoimmune parameters in CSU patients. In the era of tailored medicine, endotyping of CSU is strengthened in order to provide better treatment option for CSU patients. Authors performed important study and demonstrated the presence of serum autoantibodies to FcεRIα using their own reliable ELISA methods which were much higher in the autoimmune CSU group. Further studies are needed to validate this methods and results in the large number of CSU patients cohort.

Reviewer #2: Jang and co-workers investigate an ELISA method to detect IgG autoantibodies to the high affinity IgE receptor in patients with chronic spontaneous urticaria by using a mAb as solid phase antigen. Inhibition assays are used to assess the specificity of the method. They report that IgG to Fc�RI are present in sera of CSU patients, especially in those with an autoimmune phenotype.

GENERAL COMMENTS

A standardized in-vitro method able to detect Fc�RI autoantibodies has been sought for decades with little success. The present attempt is quite interesting, but the study shows a series of shortcomings that need to be addressed in order to increase its quality.

SPECIFIC POINTS

Line 85: Which allergens were tested?

Line 98: This is a mistake! In most cases CSU patients with thyroid autoantibodies don’t show any defect in thyroid function. Thus, testing TPO and TG auto-antibodies only in the presence of abnormal TSH levels leads to an underestimate of autoimmune study subjects.

Lines 166-7: The higher total IgE levels in CSU patients suggest autoallergy rather than IgG-mediated autoimmunity (see Schmetzer, at al. J Allergy Clin Immunol. 2018 Sep;142(3):876-882.)

Lines 168-171. See previous comment to line 98.

Line 227: Regarding the prevalence of ASST see Asero et al Clin Exp Allergy. 2001 Jul;31(7):1105-10.

Line 229: See Fusari et al. Allergy. 2005 Feb;60(2):256-8.

Lines 235-6: In most previous studies dealing with ASST and HRA, there has always been a marked discrepancy between the prevalence of these two markers, with ASST+ patients that were invariably much more frequent that HRA+ patients

Lines 251-2: The functional basophil activation tests maybe are not widely performed but remain the only reliable method to check whether the autoantibodies to the high affinity IgE receptor or to IgE are functionally active (see also Ref 17)

Lines 269-70: this is evidence against the specificity of the detection of these autoantibodies in the absence of a functional testing.

Lines 293-305: Evidence of an association between HDM and CSU is dated and poor. Atopic status as a whole prevails in auto-allergic CSU patients (type I after Gell-Cooms). See Schmetzer et al. suggested above.

Line 308: “Basophil functional assays… ”. These are missing but remain the only way to show that IgG autoantibodies are functional. Of autoimmunity

Line 309: ANA and ASST are poor surrogate markers. See also Asero et al. Eur Ann Allergy Clin Immunol. 2021 Dec 14. doi: 10.23822/

MINOR POINTS

Line 39: Why CU and not CSU?

Line 48: Reference 17 is also appropriate at this point

Line 56: a standardized method is still unavailable.

Line 114: Sera? Maybe the authors mean the YH35324 bound to the solid phase…

Line 117: dilution) were loaden onto…

Line 223: and/or IgE (delete anti- and antibodies)

Line 233: delete using.

Lines 242-3: Please delete the sentence in brackets.

Line 282: Suggestion: “Recent studies suggest that, along with mast cells, eosinophils, basophils and other cell types are involved…”

Reviewer #3: This report seeks to evaluate auto-antibodies specific for FcER1 that could be causally-linked to CSU. The approach uses a novel fusion protein in which an extracellular domain of FcER1alpha is linked to an IgG4 Fc domain. I have several concerns with the report. I found it challenging to understand the methodologies, which could relate to language, but also due to the way it is presented. Some of my concerns:

1) The main ELISA appears to actually be an inhibition assay. Why is it not described as such (noting that a separate section describes an inhibition ELISA with dose-response curve)? Why is there no data using a direct ELISA - it would likely be easier to interpret and more compelling than the current presentation? How was the inhibition concentration of 5 ug/mL chosen for the main assays, when in the dose-response study this same concentration exhibited very little inhibition?

2) Why the IgG purification in line 120, but not in the main assay? This seems like comparing apples and oranges between the approaches

3) How good are the performance characteristics of the selected cut-offs for distinguishing between CSU and HC?

4) One of the main positive findings in the paper, namely Fig 1A, shows a difference between CSU and HC that is largely explained by several outliers in the HC group with high ratios. How are these values >1 explained? Are they biologically plausible or meaningful?

5) The discussion dominates the paper and is lengthy.

Overall, there may be a story here, but the methods and results as described and shown are hard to understand and raise concerns about validity and reproducibility.

Reviewer #4: The currently accepted definition for autoimmune urticaria is finding functional autoantibodies by basophil histamine release assay or basophil activation tests. ANA and ASST do not provide sufficient evidence. Positive thyroid antibodies with low total IgE are potential surrogate markers for patients with a positive positive BHRA. Functional anti-IgE autoantibodies have also been described in chronic spontaneous urticaria. They have not been assessed in this study. Immunoassays detect functional and non-functional antibodies. Earlier work by Dr Kaplan showed that the IgG subclass is important in determining functionality in the basophil histamine release assay. Why use UAS6 - the standard clinical scoring system is over 7 days? The methodology of the ELISA is difficult to follow. The clinical utility of this test is uncertain. Some of the references are misquote, such as 7 and 8.

Reviewer #5: The manuscript “Detection of serum IgG autoantibodies to FcεRIα by ELISA in patients with chronic spontaneous urticaria” reported the results of a study that investigated CSU patients using an innovative ELISA test to detect serum Ig to FcεRIα. The study is interesting, but there are some criticism in the methods and the results are not in line with literature.

In particular:

Methods, lines 90-91. Atopy was defined when SPT showed a positive results or serum allergen specific IgE was elevated (≥0.35 IU/mL) to at least 1 common environmental allergen. Why you didn’t considered also clinical symptoms? Considering only SPT or specific IgE, you cannot diagnosed atopic dermatitis.

Methods, lines 97-98. “In cases of abnormal TSH results, thyroid autoantibodies (anti-TPO or anti-TG) were checked”. Really, there are some patients with autoimmune thyroiditis and normal TSH. Also CSU guidelines recommended the measurement of serum autoantibody IgG to thyroid peroxidase. How you checked thyroid autoantibodies only in cases of abnormal TSH?

Results, lines 176-177. “The ratio of IgG to FcεRIα was significantly lower in patients with CSU than in HCs.” This result is singular. In fact, in literature the incidence of anti- FcεRIα autoantibodies is normally higher in CSU patients (Immunobiology. 2018 Dec;223(12):807-811 - Egypt J Immunol . 2020 Jan;27(1):141-155 - J Microbiol Immunol Infect. 2020 Feb;53(1):141-147 - J Immunol Res. 2022 Feb 25;2022:6863682).

Results, lines 198-199. “The autoimmune CSU group had significantly lower ratio of IgG/IgA to FcεRIα than the non-autoimmune CSU group”. Also, this result is singular. In fact, in literature the incidence of anti- FcεRIα autoantibodies is normally higher in CSU patients with autoimmunity (especially with positive ASST).

Discussion, lines 207-208. “Higher levels of serum IgG/IgA to FcεRIα were noted in patients with CSU with autoimmune phenotypes”. In results, you state “The autoimmune CSU group had significantly lower ratio of IgG/IgA to FcεRIα than the non-autoimmune CSU group”. How can you explain this difference?

Discussion, lines 232-234. “A previous study showed higher serum IgG to FcεRIα estimated by using immunoblot assay in CSU patients, especially in those with a positive ASST, although further replication studies are not validated”. Really, some other studies confirm higher serum IgG to FcεRIα in CSU patients, as above reported.

Discussion, lines 236-240. “The discordance between previous studies and ours in the same ethnicity may be attributed to … the different detection methods applied (immune-dot assay vs ELISA). The sensitivity and specificity between these 2 different assays cannot be comparable.” In this case, the lower serum IgG to FcεRIα that you have encountered can be attributed to: (1) lower sensitivity of the method that you used; (2) lower specificity of other methods, giving false positive results. Can you demonstrate this statement?

Discussion, lines 256-257. “The present study showed a higher prevalence of IgG to FcεRIα in CSU patients than in HCs”. In results, you state “The ratio of IgG to FcεRIα was significantly lower in patients with CSU than in HCs.” How can you explain this difference?

Discussion, lines 259-260. “The autoimmune group had a higher prevalence of IgG to FcεRIα than the nonautoimmune group”. In results, you state “The autoimmune CSU group had significantly lower ratio of IgG/IgA to FcεRIα than the non-autoimmune CSU group”. How can you explain this difference?

Reviewer #6: Authors have established an ELISA for IgG, IgM and IgA autoantibodies against FceRIa, using a recombinant chimeric molecule which consists of human IgG2 and FceRIa. Using this system, the authors studied the presence of IgG, IgA and IgM autoantibodies against FceRIa in sera of patients with chronic spontaneous urticaria (CSU) and healthy controls, in relation to autoimmune characteristics and other clinical backgrounds. They detected all IgG, IgA and IgM subclasses of autoantibodies against FceRI both in sera of patients with CSU and healthy controls, but the prevalence is higher in IgG subclass of patients with CSU, especially that in patients of autoimmune group, and IgA subclass in patients with CSU of autoimmune group, but not in IgM subclass. The reason for discrepancies between this report and previous reports of the autoantibodies against FceRIa remains unclear.

Major problems:

1. The presence of autoantibodies against FceRIa is commonly compared with reactions in autologous serum skin test (ASST) and basophil functional assay (histamine release test or basophil activation test) rather than the comprehensive autoimmune characteristics defined by the authors in this study, i.e. positive results to ASST and/or antinuclear antibody (ANA) (autoimmune group). The authors stated that they were not afford to do basophils functional assays. However, the reviewer argue that the authors should show relationships of the amounts of autoantibodies and results of ASST. Authors should also describe why they employed ASST in combination to ANA, which was not taken in their previous study of the autoantibodies using dot blot assay (ref 19).

2. Authors showed that the level of IgG autoantibodies against YH35324 in patients with CSU was higher than that of healthy control. Moreover, the level in patients in autoimmune group was higher than that of non-autoimmune group. How about the comparison between patients with CSU in non-autoimmune group and healthy control?

3. For the inhibition test in ELISA, authors used YH35324 as an inhibitor. I wonder why not recombinant FceRIa, which should reflect more specific binding of autoantibodies against FceRIa.

4. Figure 1. Cutoff values for autoantibodies in this study should be shown as dotted or broken lines.

5. Page 4, line 80-84. The way of evaluation by UAS6 should be described in more detail. Is it the sum of UAS in 6 days? If so, the worst score should be 36, but sever urticaria was defined as 11 to 15 in line 83.

6. Page 7, line 132-135. These descriptions read to me that YH35324 was used as an inhibitor. If authors intend to show the superiority of their ELISA using YH35324 to ELISA using recombinant soluble FceRIa, which is the method employed in reference 10, they should use recombinant FceRIa rather than YH35324. In any case, data of such comparison is more convincing for readers. Moreover, a figure of relationship of IgG binding to HY35324 and that to recombinant FceRIa in sera of individual donors should be shown as a supplementary material.

7. Page 10, line 211. A phrase, “as well as serum free IgE in the sera of various allergic” should be deleted. Data in this manuscript do not contain any information directly support this statement.

8. Page 11, line 235-236. “However, the present study showed no ….” As pointed above, data should be shown, even if there is no difference between ASST-positive and negative groups. Table 2 showed the number and rate of ASST positive patients in IgG autoantibodies positive and negative patients, but did not show the numbers of IgG autoantibodies positive and negative patients in ASST positive and negative patients.

9. Page 260, line 260. The prevalence should also be described in the Result section as well.

10. Page 13-15, Discussions in line 276 to the end, especially to 305 are too long and tedious. The level of anti-house dust mite IgE does not have much relevance to the study of autoantibodies taken in this study.

Minor problems]

1. Page 6, line 122. Is “IgG antibody” a mistake for “anti-IgG antibody”?

2. Page 11, line 215. “to FceRIa” should be “to FceRIa or IgE”.

6. PLOS authors have the option to publish the peer review history of their article (what does this mean?). If published, this will include your full peer review and any attached files.

Reviewer #1: No

Reviewer #2: No

Reviewer #3: No

Reviewer #4: No

Reviewer #5: No

Reviewer #6: **Yes: **Michihiro Hide

---

## [Author Response · Author response to Decision Letter 0]

28 Jul 2022

Reviewer #1: The objective of this study was to investigate the prevalence of circulating autoantibodies to FcεRIα in association with clinical/autoimmune parameters in CSU patients. In the era of tailored medicine, endotyping of CSU is strengthened in order to provide better treatment option for CSU patients. Authors performed important study and demonstrated the presence of serum autoantibodies to FcεRIα using their own reliable ELISA methods which were much higher in the autoimmune CSU group. Further studies are needed to validate this methods and results in the large number of CSU patients cohort.

Our response: We thank the reviewer for the positive comment on this study. Indeed, this study highlighted the presence of autoantibodies to FcεRIα, and compared it between the autoimmune and non-autoimmune groups in patients with CSU, which should be further validated in larger cohorts of CU patients. 

Reviewer #2: Jang and co-workers investigate an ELISA method to detect IgG autoantibodies to the high affinity IgE receptor in patients with chronic spontaneous urticaria by using a mAb as solid phase antigen. Inhibition assays are used to assess the specificity of the method. They report that IgG to Fc�RI are present in sera of CSU patients, especially in those with an autoimmune phenotype.

GENERAL COMMENTS

A standardized in-vitro method able to detect Fc�RI autoantibodies has been sought for decades with little success. The present attempt is quite interesting, but the study shows a series of shortcomings that need to be addressed in order to increase its quality.

SPECIFIC POINTS

Line 85: Which allergens were tested?

Our response: We evaluated the atopic status of enrolled patients using skin prick tests with common aeroallergens including Dermatophagoides pteronyssinus, D. farinae, cat, dog, cockroach, tree pollen mixture, grass pollen mixture, mugwort, ragweed, Humulus japonicus, Aspergillus, and Alternaria which are the most prevalent inhalant allergens in our environment.

Line 98: This is a mistake! In most cases CSU patients with thyroid autoantibodies don’t show any defect in thyroid function. Thus, testing TPO and TG auto-antibodies only in the presence of abnormal TSH levels leads to an underestimate of autoimmune study subjects.

Our response: We completely agree with your comments. In this study we used the laboratory test results retrospectively collected from the enrolled patients. In fact, the healthcare system of our country recommends that testing TPO and TG auto-antibodies would be preceded by the abnormal thyroid function test results. In this study, thyroid autoantibodies were measured only if the patients were indicated. Therefore, with this limitation, we did not use thyroid autoantibodies to define the autoimmune group of CSU. 

Lines 166-7: The higher total IgE levels in CSU patients suggest autoallergy rather than IgG-mediated autoimmunity (see Schmetzer, at al. J Allergy Clin Immunol. 2018 Sep;142(3):876-882.)

Our response: Thank you for your comments. Indeed, higher levels of serum total IgE were reported to be more related to autoallergy in CSU than in type IIb CSU in previous studies. Consequently, higher total IgE levels in patients with CSU may result in the lower prevalence of autoantibodies to FcεRIα in this study. However, CSU patients having serum autoantibodies to FcεRIα had higher median value of serum total IgE levels than that of HCs (219.0 [68.0;525.0] vs. 46.1[22.0;97.9], P value 0.004) in this study. Therefore, higher total IgE levels in CSU patients does not always rule out the IgG-mediated autoimmunity in CSU patients in this cohort. 

Lines 168-171. See previous comment to line 98.

Our response: Thank you for your comments. As we demonstrated, we evaluated thyroid autoimmunity in this study with the limitation of the healthcare system. Therefore, we evaluated thyroid autoimmunity in the patients if they were indicated.

Line 227: Regarding the prevalence of ASST see Asero et al Clin Exp Allergy. 2001 Jul;31(7):1105-10.

Our response: Thank you for your comments. We added the reference which you mentioned in the manuscript. (L 232) 

Line 229: See Fusari et al. Allergy. 2005 Feb;60(2):256-8.

Our response: Thank you for your comments. We added the reference which you mentioned in the manuscript as following: Autoimmune thyroiditis was reported to be related to the persistent positive result of ASST in patients with CSU independent of urticaria symtpoms, questioning the role of ASST in the CSU patients with autoimmune thyroiditis. (L 233:234)

Lines 235-6: In most previous studies dealing with ASST and HRA, there has always been a marked discrepancy between the prevalence of these two markers, with ASST+ patients that were invariably much more frequent that HRA+ patients

Our response: Thank you for raising this point. This discrepancy may basically come from the methodological difference between these two methods. Each test could cover different spectrum of CSU with autoimmunity. A recent review about ASST in CSU demonstrated that almost all the patients with the positive result of basophil functional test including HRA were also ASST-positive, whereas a partial of ASST-positive patients were also basophil functional test positive. Circulating histaminergic factors in addition to autoantibodies were pointed as the potential reason for the auto-reactivity of ASST-positive patients. The positive result of ASST is a poor predictor of a positive basophil functional test, but ASST-negative patients could be excluded for the presence of functional circulating autoantibodies detectable by HRA based on high negative predictive value of ASST result. 

Lines 251-2: The functional basophil activation tests maybe are not widely performed but remain the only reliable method to check whether the autoantibodies to the high affinity IgE receptor or to IgE are functionally active (see also Ref 17)

Our response: We entirely agree with your comments. It is the fact that basophil histamine release assay and basophil activation test are reliable methods to evaluate functional autoantibodies to FcεRIα in autoimmune CSU. However, the basophil function test must need considerable amount of samples, and it was not enough for the present study to evaluate the basophil function test considering the amount of stored patient samples. The present study aimed to evaluate the presence of serum autoantibodies to FcεRIα in CSU using YH35324. Further studies should be followed to validate the functional effect of these autoantibodies on basophils. 

Lines 269-70: this is evidence against the specificity of the detection of these autoantibodies in the absence of a functional testing.

Our response: Thank you for your comments. We agree with your comments. However, in order to confirm its binding specificity, we did IgG-ELISA (competitive) inhibition test with serial additions of YH35324 as shown in Supplementary Figure 2. Significant inhibitions were noted with increasing doses of YH35324 preincubated in dose-dependent manners, while no significant inhibition was noted with addition of high dose of house dust mite allergen, indicating that circulating IgG autoantibodies detected in sera using our ELISA have binding specificity to FcεRIα, although we could not confirm their function whether they could release histamine from basophils or not. 

Lines 293-305: Evidence of an association between HDM and CSU is dated and poor. Atopic status as a whole prevails in auto-allergic CSU patients (type I after Gell-Cooms). See Schmetzer et al. suggested above.

Our response: Thank you for your comments. We agree with your comments. Therefore, we have deleted the section describing the association between HDM and CSU as suggested. 

Line 308: “Basophil functional assays… ”. These are missing but remain the only way to show that IgG autoantibodies are functional. Of autoimmunity

Our response: Thank you for your comments. We added as suggested. (L304:L305) 

Line 309: ANA and ASST are poor surrogate markers. See also Asero et al. Eur Ann Allergy Clin Immunol. 2021 Dec 14. doi: 10.23822/

Our response: Thank you for your comments. We admitted that the absence of the basophil function assays is critical limitation of this study. However, as we mentioned, this kind of test is not yet appropriate for performing in outpatient clinic. It is easier to do ASST or ANA test for screening autoimmunity in CSU patients than doing basophil functional test considering high negative predictive value of ASST, ranging from 59% to 100%. We added the reference as suggested. (L 306:311)

MINOR POINTS

Line 39: Why CU and not CSU?

Our response: Thank you for your comments. We have changed from the word “CU” to CSU through the manuscript. (L39)

Line 48: Reference 17 is also appropriate at this point

Our response: We have modified as suggested. (L48)

Line 56: a standardized method is still unavailable.

Our response: We have modified as suggested. (L56)

Line 114: Sera? Maybe the authors mean the YH35324 bound to the solid phase…

Our response: Thank you for your comments. We have changed the word “sera” to plate. (L114)

Line 117: dilution) were loaden onto…

Our response: We have modified as suggested. (L117) … 

Line 223: and/or IgE (delete anti- and antibodies)

Our response: We have modified as suggested. (L239) … 

Line 233: delete using.

Our response: Thank you for your comments. We have removed using in the revised manuscript as suggested. (L241)

Lines 242-3: Please delete the sentence in brackets.

Our response: Thank you for your comments. We have deleted the sentence as suggested. (L 250)

Line 282: Suggestion: “Recent studies suggest that, along with mast cells, eosinophils, basophils and other cell types are involved…”

Our response: We have modified as suggested. (L289:L290)

Reviewer #3: This report seeks to evaluate auto-antibodies specific for FcER1 that could be causally-linked to CSU. The approach uses a novel fusion protein in which an extracellular domain of FcER1alpha is linked to an IgG4 Fc domain. I have several concerns with the report. I found it challenging to understand the methodologies, which could relate to language, but also due to the way it is presented. Some of my concerns:

1) The main ELISA appears to actually be an inhibition assay. Why is it not described as such (noting that a separate section describes an inhibition ELISA with dose-response curve)? 

Our response: Thank you for your comments. We demonstrated the result of ELISA inhibition test with serial additions of YH35324 in the supplementary Figure 2. 

Why is there no data using a direct ELISA - it would likely be easier to interpret and more compelling than the current presentation? 

Our response: We already performed a direct ELISA with YH35324, but there was much background activity of non-specific IgG bindings as each serum have very high IgG level. Therefore, we compared the ratio of inhibition could be better for lowering the background signals of the measurement as previously reported by European investigators (Allergy 2020;75:3208 by Maurer M et al.) 

How was the inhibition concentration of 5 ug/mL chosen for the main assays, when in the dose-response study this same concentration exhibited very little inhibition?

Our response: Thank you for your comments. We did preliminary experiments to determine the optimal concentration of YH35324 to determine the presence of IgG autoantibody level compared to controls and found that 5 µg/mL is the optimal concentration to differentiate positive and negative groups in each serum.

2) Why the IgG purification in line 120, but not in the main assay? This seems like comparing apples and oranges between the approaches

Our response: Thank you for your comments. We applied IgG purification for the preparing measurement of IgG to FcεRIα to reduce non-specific binding to YH35324. In case of measurement of IgA and IgM, the levels of IgA and IgM were so low for using purification process. 

3) How good are the performance characteristics of the selected cut-offs for distinguishing between CSU and HC? 

Our response: The sensitivity and the specificity are 8.14% and 85.71%. The cut-off values for the presence of autoantibodies in patients with CSU were determined at the 5% percentile ratio of non-autoimmune CSU. Therefore, it is not appropriate using this cut-off values for distinguishing between CSU and HC. The sensitivity and the specificity for determining the autoimmune CSU and non-autoimmune CSU groups are 15.63% and 96.30% respectively. 

4) One of the main positive findings in the paper, namely Fig 1A, shows a difference between CSU and HC that is largely explained by several outliers in the HC group with high ratios. How are these values >1 explained? Are they biologically plausible or meaningful? 

Our response: We thank the reviewer for raising this point. We do also have been concerned about the value over 1. However, it has been concluded that they might be derived from non-specific binding effects in this assay.

5) The discussion dominates the paper and is lengthy. 

Our response: Thank you for your comments. We have removed the part about relationship between HDM sensitization and CSU. The revised manuscript appears tidier. 

Overall, there may be a story here, but the methods and results as described and shown are hard to understand and raise concerns about validity and reproducibility.

Our response: Thank you for your comments. We did our best to trim the manuscript as suggested.

Reviewer #4: The currently accepted definition for autoimmune urticaria is finding functional autoantibodies by basophil histamine release assay or basophil activation tests. ANA and ASST do not provide sufficient evidence. Positive thyroid antibodies with low total IgE are potential surrogate markers for patients with a positive positive BHRA. 

Our response: Thank you for your comments. Indeed, the absence of the basophil functional test is critical limitation of this study. We aimed to validate new measurement of serum autoantibodies to FcεRIα in CSU patients. The currently accepted definition for type IIb autoimmune CSU needs the presence of IgG anti-FcεRIα which means a signal reduction by more than 50% and it is not clearly standardized. The ASST with a high negative predictive value ranging from 59% to 100% could be used to exclude the presence of functional circulating autoantibodies to FcεRIα in CSU patients. In this study, we used ANA and ASST result to define autoimmune CSU and non-autoimmune CSU for validating the new measurement of serum autoantibodies. In addition, CSU patients with positive thyroid antibodies did not have autoantibodies to FcεRIα independent of serum total IgE levels in our cohort. 

Functional anti-IgE autoantibodies have also been described in chronic spontaneous urticaria. They have not been assessed in this study. 

Our response: We focused on the measurement of autoantibodies to FcεRIα in CSU patients. However, we evaluated IgG to IgE autoantibodies in this cohort, which showed no significant difference. (Data were not shown) 

Immunoassays detect functional and non-functional antibodies. 

Our response: Thank you for your comments. We admitted the absence of evaluating functional antibodies was the limitation of this study. 

Earlier work by Dr Kaplan showed that the IgG subclass is important in determining functionality in the basophil histamine release assay.

Our response: Unfortunately, we did not evaluate the IgG subclass of autoantibodies to FcεRIα. 

Why use UAS6 - the standard clinical scoring system is over 7 days? 

Our response: Thank you for your comments. We have replaced UAS with UAS-15 and removed the word “UAS6” to descript the scoring method more precisely. (L80)

The methodology of the ELISA is difficult to follow. The clinical utility of this test is uncertain.

Our response: The measurement of serum autoantibodies to FcεRI in CSU patients has not yet been standardized. We aimed to demonstrate the presence of serum autoantibodies using ELISA with YH35324 in our cohort and evaluated its functional relevance according to autoimmune status (based on the results of ASST and ANA). We used the same ELISA method published by Maurer M et al. (Allergy 2020;75:3208). Further replication studies are needed in other cohorts in other regions.

Some of the references are misquote, such as 7 and 8. 

Our response: Thank you for your comments. We have modified the quotation of the references. 

Reviewer #5: The manuscript “Detection of serum IgG autoantibodies to FcεRIα by ELISA in patients with chronic spontaneous urticaria” reported the results of a study that investigated CSU patients using an innovative ELISA test to detect serum Ig to FcεRIα. The study is interesting, but there are some criticism in the methods and the results are not in line with literature.

In particular:

Methods, lines 90-91. Atopy was defined when SPT showed a positive results or serum allergen specific IgE was elevated (≥0.35 IU/mL) to at least 1 common environmental allergen. Why you didn’t considered also clinical symptoms? Considering only SPT or specific IgE, you cannot diagnosed atopic dermatitis. 

Our response: Thank you for your comments. In fact, the word “atopy” does not mean the disease as atopic dermatitis in this context. We aimed to define the atopic tendency of the enrolled patients as “atopy”. There is no need to consider clinical symptoms to define atopic tendency. 

Methods, lines 97-98. “In cases of abnormal TSH results, thyroid autoantibodies (anti-TPO or anti-TG) were checked”. Really, there are some patients with autoimmune thyroiditis and normal TSH. Also CSU guidelines recommended the measurement of serum autoantibody IgG to thyroid peroxidase. How you checked thyroid autoantibodies only in cases of abnormal TSH? 

Our response: Thank you for your comments. We could not screen the presence of serum autoantibodies to thyroglobulin (TG) or thyroid-specific peroxidase (TPO) antigens in all study subjects with CSU enrolled in this study. The measurement of autoantibodies related to thyroid disease is recommended to be followed by abnormal thyroid function test results (including high TSH level) following the health care system guideline in this country. Therefore, we limitedly evaluated thyroid autoantibodies. 

Results, lines 176-177. “The ratio of IgG to FcεRIα was significantly lower in patients with CSU than in HCs.” This result is singular. In fact, in literature the incidence of anti- FcεRIα autoantibodies is normally higher in CSU patients (Immunobiology. 2018 Dec;223(12):807-811 - Egypt J Immunol . 2020 Jan;27(1):141-155 - J Microbiol Immunol Infect. 2020 Feb;53(1):141-147 - J Immunol Res. 2022 Feb 25;2022:6863682). 

Our response: Thank you for your comments. In this study, the ratio was defined as YH35324 preincubated to mock-preincubated values. Theoretically, preincubated with YH35324 could diminish the concentration of autoantibodies to FcεRIα. Therefore, in this study, the lower ratio of IgG to FcεRIα indicates higher levels of anti-FcεRIα IgG which was described in the text and published by Maurer M et al. (Allergy 2020; 75:3208).

Results, lines 198-199. “The autoimmune CSU group had significantly lower ratio of IgG/IgA to FcεRIα than the non-autoimmune CSU group”. Also, this result is singular. In fact, in literature the incidence of anti- FcεRIα autoantibodies is normally higher in CSU patients with autoimmunity (especially with positive ASST). Discussion, lines 207-208. “Higher levels of serum IgG/IgA to FcεRIα were noted in patients with CSU with autoimmune phenotypes”. In results, you state “The autoimmune CSU group had significantly lower ratio of IgG/IgA to FcεRIα than the non-autoimmune CSU group”. How can you explain this difference?

Our response: Thank you for your comments. The lower ratio of YH35324 preincubated to mock-preincubated values demonstrated the higher concentration of autoantibodies measured by YH35324. This result is consistent with previous studies (Allergy 2020; 75:3208, and Figure 1 in the present study), therefore increased circulating autoantibody to FcεRIα is associated with the autoimmune phenotype of CU (based on results of ASST and ANA). 

Discussion, lines 232-234. “A previous study showed higher serum IgG to FcεRIα estimated by using immunoblot assay in CSU patients, especially in those with a positive ASST, although further replication studies are not validated”. Really, some other studies confirm higher serum IgG to FcεRIα in CSU patients, as above reported. 

Our response: Thank you for your comments. The lower ratio of IgG to FcεRIα indicate s higher level of circulating IgG to FcεRIα, therefore, the result of the present study is comparable with previous reports showing higher prevalence of serum autoantibodies, IgG to FcεRIα in CSU patients (Allergy 2020; 75:3208).

Discussion, lines 236-240. “The discordance between previous studies and ours in the same ethnicity may be attributed to … the different detection methods applied (immune-dot assay vs ELISA). The sensitivity and specificity between these 2 different assays cannot be comparable.” In this case, the lower serum IgG to FcεRIα that you have encountered can be attributed to: (1) lower sensitivity of the method that you used; (2) lower specificity of other methods, giving false positive results. Can you demonstrate this statement? 

Our response: The discordance between the previous study and the present study was that the present study showed no significant difference in IgG to FcεRIα between the ASST-positive and ASST-negative group even in the same ethnicity. The previous study compared nearly the same number of patients according to the ASST results (64 ASST-positive and 61 ASST-negative patients) compared to relatively small number of CSU patients having positive ASST results (n=20) in the present study. Considering various positive rates of ASST results, we could not compare sensitivity and specificity of these two different studies. 

Discussion, lines 256-257. “The present study showed a higher prevalence of IgG to FcεRIα in CSU patients than in HCs”. In results, you state “The ratio of IgG to FcεRIα was significantly lower in patients with CSU than in HCs.” How can you explain this difference? 

Our response: Thank you for your comments. As we mentioned, the lower ratio implied the higher concentration of autoantibodies FcεRIα in CSU patients. Therefore, the lower ratio of IgG to FcεRIα in CSU patients could be translated into a higher prevalence of IgG to FcεRIα in CSU than in HCs.

Discussion, lines 259-260. “The autoimmune group had a higher prevalence of IgG to FcεRIα than the nonautoimmune group”. In results, you state “The autoimmune CSU group had significantly lower ratio of IgG/IgA to FcεRIα than the non-autoimmune CSU group”. How can you explain this difference? 

Our response: Thank you for your comments. The lower ratio of IgG to FcεRIα in the autoimmune CSU group (than in the non-autoimmune CSU group) could imply a higher concentration of IgG to FcεRIα, which can be translated into a higher prevalence of IgG to FcεRIα in the autoimmune CSU group.

Reviewer #6: Authors have established an ELISA for IgG, IgM and IgA autoantibodies against FceRIa, using a recombinant chimeric molecule which consists of human IgG2 and FceRIa. Using this system, the authors studied the presence of IgG, IgA and IgM autoantibodies against FceRIa in sera of patients with chronic spontaneous urticaria (CSU) and healthy controls, in relation to autoimmune characteristics and other clinical backgrounds. They detected all IgG, IgA and IgM subclasses of autoantibodies against FceRI both in sera of patients with CSU and healthy controls, but the prevalence is higher in IgG subclass of patients with CSU, especially that in patients of autoimmune group, and IgA subclass in patients with CSU of autoimmune group, but not in IgM subclass. The reason for discrepancies between this report and previous reports of the autoantibodies against FceRIa remains unclear.

Major problems:

1. The presence of autoantibodies against FceRIa is commonly compared with reactions in autologous serum skin test (ASST) and basophil functional assay (histamine release test or basophil activation test) rather than the comprehensive autoimmune characteristics defined by the authors in this study, i.e. positive results to ASST and/or antinuclear antibody (ANA) (autoimmune group). The authors stated that they were not afford to do basophils functional assays. However, the reviewer argue that the authors should show relationships of the amounts of autoantibodies and results of ASST. Authors should also describe why they employed ASST in combination to ANA, which was not taken in their previous study of the autoantibodies using dot blot assay (ref 19). 

Our response: Thank you for your comments. Data were not shown, but we mentioned that the ratio of IgG to FcεRIα had no significant difference comparing between the ASST-positive and ASST-negative group. For this reason, we combined ANA result to ASST to define the non-autoimmune group in CSU who might have low prevalence of serum autoantibodies. The previous study sharing same ethnicity instead compared ASST-positive and ASST-negative patients with nearly same number of patients. Unfortunately, we could not evaluate functional effects of IgG to FcεRIα detected in this study.

2. Authors showed that the level of IgG autoantibodies against YH35324 in patients with CSU was higher than that of healthy control. Moreover, the level in patients in autoimmune group was higher than that of non-autoimmune group. How about the comparison between patients with CSU in non-autoimmune group and healthy control? 

Our response: Thank you for your comments. Patients with CSU in the non-autoimmune group tended to show lower ratio of IgG autoantibody to FcεRIα than that of HCs, however, no statistical significance was noted. The ratio of IgA/IgM to FcεRIα showed no difference between the 2 groups.

3. For the inhibition test in ELISA, authors used YH35324 as an inhibitor. I wonder why not recombinant FceRIa, which should reflect more specific binding of autoantibodies against FceRIa. 

Our response: We evaluated the ratio of IgG to FcεRIα using recombinant FcεRIα (as a solid phase antigen) and showed no significant differences between CSU patients and HCs, therefore we did not use recombinant FcεRIα as an inhibitor for ELISA inhibition test.

4. Figure 1. Cutoff values for autoantibodies in this study should be shown as dotted or broken lines. 

Our response: Thank you for your comments. We added cutoff values of each autoantibody to FcεRIα as dotted lines in both Figure 1 and 2.

5. Page 4, line 80-84. The way of evaluation by UAS6 should be described in more detail. Is it the sum of UAS in 6 days? If so, the worst score should be 36, but sever urticaria was defined as 11 to 15 in line 83. 

Our response: Thank you for your comments. We evaluated disease activity using the 15-point urticaria activity score (UAS-15). UAS-15 includes degree of pruritus and the number/size/distribution/duration of wheal during the preceding week. Each parameter is scored from 0 to 3 and the maximum UAS-15 with total five parameters can be 15, which were revised in the text.

6. Page 7, line 132-135. These descriptions read to me that YH35324 was used as an inhibitor. If authors intend to show the superiority of their ELISA using YH35324 to ELISA using recombinant soluble FceRIa, which is the method employed in reference 10, they should use recombinant FceRIa rather than YH35324. In any case, data of such comparison is more convincing for readers. Moreover, a figure of relationship of IgG binding to HY35324 and that to recombinant FceRIa in sera of individual donors should be shown as a supplementary material. 

Our response: Thank you for your comments. We aimed to show compare the efficacy of ELISA with YH35324 and recombinant FcεRIα, and do ELISA under the same protocol, except only replacing YH35324 with recombinant FcεRIα. The recombinant FcεRIα was used as an inhibitor measuring IgG binding to recombinant FcεRIα. The manuscript has been edited as suggested. (L 136) And we added Supplementary Figure 1 showing the correlation between IgG binding to YH35324 and that to recombinant FcεRIα. The manuscript has been edited as suggested. (L 172:L173) 

7. Page 10, line 211. A phrase, “as well as serum free IgE in the sera of various allergic” should be deleted. Data in this manuscript do not contain any information directly support this statement. 

Our response: Thank you for your comments. The phrase has been deleted. 

8. Page 11, line 235-236. “However, the present study showed no ….” As pointed above, data should be shown, even if there is no difference between ASST-positive and negative groups. Table 2 showed the number and rate of ASST positive patients in IgG autoantibodies positive and negative patients, but did not show the numbers of IgG autoantibodies positive and negative patients in ASST positive and negative patients. 

Our response: Thank you for your comments. We attached the supplementary figure S3 comparing autoantibodies between ASST positive and negative groups in CSU. (L205:L207)

9. Page 260, line 260. The prevalence should also be described in the Result section as well. 

Our response: Thank you for your comments. We additionally demonstrated the prevalence of autoantibodies to FcεRIα in patients with CSU in the Result section and Table 2.

 (L187) 

10. Page 13-15, Discussions in line 276 to the end, especially to 305 are too long and tedious. The level of anti-house dust mite IgE does not have much relevance to the study of autoantibodies taken in this study. 

Our response: Thank you for your comments. We have removed the section describing the relationship between house dust mite and CSU in the text. 

Minor problems]

1. Page 6, line 122. Is “IgG antibody” a mistake for “anti-IgG antibody”? 

Our response: Thank you for your comments. We have changed as suggested. (L122) 

2. Page 11, line 215. “to FceRIa” should be “to FceRIa or IgE”.

Our response: Thank you for your comments. We have changed as suggested. (L221)

---

## [Editor Report · Decision Letter 1]

9 Aug 2022

Detection of serum IgG autoantibodies to FcεRIα by ELISA in patients with chronic spontaneous urticaria

PONE-D-22-12472R1

Dear Dr. Park,

We’re pleased to inform you that your manuscript has been judged scientifically suitable for publication and will be formally accepted for publication once it meets all outstanding technical requirements.

Kind regards,

Cheorl-Ho Kim, Ph.D.

Academic Editor

PLOS ONE

Additional Editor Comments (optional):

August 9, 2022

Dear Dr Park,

Thanks for your submission to Plos One.

I have checked your revision and found it valuable for publication, although there are still controversial issues on the clinical tests or examinations.

As you know, the serum IgG autoantibodies to FcεRIα has been detected by other works before your submisison to our Plos One.

However, I recognized your work as a systemic validation from patients with chronic spontaneous urticaria.

Thank you

Sincerely

Cheorl-Ho Kim PhD

Academic Editor

Plos One
---

## [Editor Report · Acceptance letter]

11 Aug 2022

PONE-D-22-12472R1 

Detection of serum IgG autoantibodies to FcεRIα by ELISA in patients with chronic spontaneous urticaria 

Dear Dr. Park:

I'm pleased to inform you that your manuscript has been deemed suitable for publication in PLOS ONE. Congratulations! Your manuscript is now with our production department. 

Kind regards, 

on behalf of

Professor Cheorl-Ho Kim 

Academic Editor

PLOS ONE